# Circulating Neuroendocrine Tumor Biomarkers: Past, Present and Future

**DOI:** 10.3390/jcm11195542

**Published:** 2022-09-21

**Authors:** Paweł Komarnicki, Jan Musiałkiewicz, Alicja Stańska, Adam Maciejewski, Paweł Gut, George Mastorakos, Marek Ruchała

**Affiliations:** 1Department of Endocrinology, Metabolism and Internal Diseases, Poznan University of Medical Sciences, 60-355 Poznań, Poland; 2Unit of Endocrinology, Diabetes Mellitus and Metabolism, Aretaieion University Hospital, Medical School, National and Kapodistrian University of Athens, 157 72 Athens, Greece

**Keywords:** neuroendocrine tumors, biomarkers, chromogranin A, neuroendocrinology, neuroendocrine neoplasms, microRNA, NETest, 5-HIAA

## Abstract

Neuroendocrine tumors are a heterogeneous group of neoplasms originating from the diffuse endocrine system. Depending on primary location and hormonal status, they range in terms of clinical presentation, prognosis and treatment. Functional tumors often develop symptoms indicating an excess of hormones produced by the neoplasm (exempli gratia insulinoma, glucagonoma and VIPoma) and can be diagnosed using monoanalytes. For non-functional tumors (inactive or producing insignificant amounts of hormones), universal biomarkers have not been established. The matter remains an important unmet need in the field of neuroendocrine tumors. Substances researched over the years, such as chromogranin A and neuron-specific enolase, lack the desired sensitivity and specificity. In recent years, the potential use of Circulating Tumor Cells or multianalytes such as a circulating microRNA and NETest have been widely discussed. They offer superior diagnostic parameters in comparison to traditional biomarkers and depict disease status in a more comprehensive way. Despite a lot of promise, no international standards have yet been developed regarding their routine use and clinical application. In this literature review, we describe the analytes used over the years and cover novel biomarkers that could find a use in the future. We discuss their pros and cons while showcasing recent advances in the field of neuroendocrine tumor biomarkers.

## 1. Introduction

Neuroendocrine tumors (NETs) are a diverse group of neoplasms. They are made from diffuse neuroendocrine system cells, which are present throughout the human body. The prevalence of neuroendocrine tumors ranges between 2.5 and 8.35 cases per 10,000, with incidence rates rising in recent years [1,2,3]. NETs fulfill the rare disease criteria according to the Orphan Drug Act (a condition affecting less than 200,000 people in the United States). Neuroendocrine Tumors, along with a second subunit, Neuroendocrine Carcinomas (NECs), are a part of a group named Neuroendocrine Neoplasms (NENs), as per WHO nomenclature [4]. Despite a similar origin from neuroendocrine tissue, both of them have their own distinct morphological features and genomic signatures. NETs can be both low- and high-grade, whereas NEC are high-grade by definition. In order to distinguish NETs from NECs, pathologists utilize tissue biomarkers of neuroendocrine lineage such as synaptophysin, chromogranin A and somatostatin receptors, some of which can also be used as circulating biomarkers [5]. Due to significant differences between both groups in terms of clinical presentation, applicable biomarkers and the natural course of the disease, this review focuses mainly on NETs. Depending on their embryonic origin (from which part of the primary gut tube the tumor originates from), NETs can be divided into three groups: foregut, midgut and hindgut, each with their own distinct characteristics [6]. The primary location of the tumor, and which part of the primary gut tube the neoplasm stems from, affects the application and clinical utility of different biomarkers. For instance, 5-hydroxyindoleacetic acid (5-HIAA) shows a higher sensitivity in midgut NENs than in pancreatic NENs, whereas the expression of chromogranin A (CgA) is lower in hindgut tumors, compared to midgut and foregut [7]. Biomarkers aside, embryonic origin directly affects the diagnostic and treatment procedures, as patients diagnosed with foregut NENs should undergo screening for MEN-1 syndrome [8]. Table 1 presents examples of primary tumor locations falling into each category.

NETs can be divided based on their ability to release hormones (functional tumors) or not (non-functional tumors). Neoplasms producing clinically insignificant amounts of hormones also fall into the latter bracket. In the case of functional tumors, hormones released into the circulation allow for utilizing them as biomarkers, as shown on Table 2. 

Additionally, excess concentration of a given hormone is linked with symptoms specific to the disease. For example, insulinoma, an insulin-producing tumor most commonly found in the pancreas, typically presents with hypoglycemic episodes [10]. These characteristics allow for a relatively quick and accurate diagnosis, however, there are certain limitations. Functional tumors constitute a minority of all NENs (10–40%), with some of them being extremely rare (<100 cases described worldwide) [8]. Clinical manifestations may change over the course of the disease and there are a number of factors that cause similar symptoms or test results (for instance, exogenous insulin intake or Hirata’s disease mimic insulinoma). Hormonal testing should be guided by the presence of symptoms in an individual; screening for the disease in patients with asymptomatic disease isn’t required [11]. On the contrary, non-functional tumors lack a specific biomarker or the spectrum of symptoms that would allow for a quick diagnosis. The patient might not spot any manifestations of the disease until the lesion starts infiltrating nearby tissue or the metastases start impairing the function of distant organs. In fact, 12–22% of patients present at the metastatic stage, despite the slow growth of neuroendocrine tumors [3].

Over the years, researchers and physicians tried to find molecules that could help diagnose neuroendocrine tumors early, improving patient prognosis. Due to the heterogeneous nature of neuroendocrine tumors, the search for a one-for-all analyte has so far been unsuccessful. This article aims to review circulating biomarkers used in daily clinical practice over the years, as well as to discuss the latest findings regarding the potential future biomarkers.

## 2. Materials and Methods

Upon forming the topic of the review, a thorough literature search was conducted. Initially, the guidelines of selected endocrinological societies were analyzed (Polish Society of Endocrinology, Polish Network of Neuroendocrine Tumors, European Society for Medical Oncology, European Neuroendocrine Tumor Society, North American Neuroendocrine Tumor Society). Afterwards, the PubMed database was searched using general terms: “neuroendocrine tumors”, “neuroendocrine neoplasms”, “neuroendocrine tumor biomarkers”, “neuroendocrine neoplasms biomarkers”, “NET biomarkers”, “NEN biomarkers” and “neuroendocrine biomarkers”. A second detailed search was conducted after a review of the initial results, focusing on the substances that showed the most merit in the guidelines and analyzed papers. Terms screened for included: “chromogranin A”, “chromogranin B”, “granins”, “5-hydroxyindoleacetic acid”, “5-HIAA”, “neuron-specific enolase”, “NSE”, “NETest”, “microRNA”, “Circulating tumor cells” and “CTC”, as well as variations of the searches above combining the terms with the words “biomarkers”, “NET” and “neuroendocrine”. The alternate spelling of certain words was accounted for (tumor/tumour, neurospecific/neuro-specific/neuron-specific). Based on the results of the searches mentioned above, a manuscript was drafted. When citing original research, the number of patients involved and methodology was taken into account. In certain topics (namely CTC and miRNA), the number of published original papers remains low because of their novel status and recent discoveries, presenting a limitation of the review. After the verification of search results, titles and abstracts, a thorough analysis of 265 selected papers was conducted. The reference lists of selected papers were also analyzed and 25 additional relevant articles were found. In total, 163 papers were chosen for the review. Included in the total number were 6 additional articles suggested by the reviewers after the first round of peer-review and 7 abstracts from the 19th Annual ENETS conference.

## 3. Discussion

### 3.1. Granins

In 1967, Blaschko et al. described the soluble proteins found in bovine secretory granules, which they named chromogranins [12]. Some notable members of that group, discovered in later years, include chromogranins A (CgA) and B (CgB, also called secretogranin I), and secretogranins II (which used to be called chromogranin C), III and IV [13]. Since their discovery, numerous articles have been published describing their role in neuroendocrine secretion [14,15,16,17]. Elevated bodily fluid concentrations of different granins (most notably CgA) among patients with hormonally active neoplasms have been some of the most important observations established in that research and with far-reaching clinical implications. Subsequently, their role as a potential biomarker of hormonally active neoplasms (e.g., NETs, pheochromocytoma, medullary thyroid cancer and pituitary gland tumors) has been analyzed [18,19,20].

#### 3.1.1. Chromogranin A

Ever since its discovery over 50 years ago, chromogranin A, a hydrophilic glycoprotein made up of 439 amino acids, remains the most widely used NET biomarker in clinical practice [21]. It is present in most neuroendocrine cells, as well as in neuroendocrine tumor cells, most notably midgut and pancreatic neoplasms [22]. It has been a staple in NET diagnostics over the years, as noted by the guidelines from numerous scientific societies: European Neuroendocrine Tumor Society (ENETS) [23], North American Neuroendocrine Tumor Society (NANETS) [24], Polish Network of Neuroendocrine Tumours [8]. CgA concentration correlates with tumor burden; the highest values are observed in metastatic NETs [24]. Depending on the type of the tumor and location, sensitivity and specificity range between 68–81% and 56–100%, respectively [25,26,27]. Similarly to 5-HIAA, its sensitivity and specificity differs depending on the location; midgut tumors express CgA most often, foregut and hindgut less so [7]. Nobels et al. demonstrated that an elevated CgA is a valuable marker in patients with gastrinomas, pheochromocytomas, carcinoid tumors and non-functioning pancreatic NETs. Elevated CgA levels were found in 100%, 89%, 80% and 69%, respectively [28]. A high sensitivity of CgA in gastrinoma makes it useful for a post-treatment follow-up [29]. Additionally, CgA showcases a greater utility in monitoring the progression of the disease and treatment response than as a diagnostic biomarker, as revealed by a 2018 meta-analysis on the subject, and increased values of CgA can predate radiological progression or tumor recurrence [30,31,32]. Recent meta-analysis of bronchopulmonary Neuroendocrine Neoplasms (bpNEN) showed sensitivity of as little as 35%, with 94% specificity [33,34]. Moreover, CgA concentration correlates with tumor burden; the highest values are observed in metastatic NETs [24], in which the specificity of 100% and sensitivity between 78 and 80% have been reported [25]. The 2015 ENETS guidelines noted the lack of systematic empirical evidence for use of CgA in bpNEN [35]. In the wake of recent research, current guidelines state that treatment decisions should not be based solely on CgA results [11].

Despite relatively good sensitivity and specificity in certain tumors, CgA has some flaws. There are no standards available regarding testing and there are significant differences between the available assays (CgA can be measured in plasma and serum, using ELISA, IRMA and RIA methods). It is therefore recommended to use the same test (preferably in the same laboratory), when comparing results [8,36]. It is noteworthy that several factors might influence CgA concentration. Most common conditions include atrophic gastritis, Helicobacter pylori infection, kidney failure, liver cirrhosis, inflammatory bowel diseases, and other non-neuroendocrine neoplasms [37,38]. Additionally, certain medications may cause false-positive results by increasing gastrin secretion, namely proton pump inhibitors and H2-receptor antagonists [39,40,41]. In order to adequately evaluate CgA level, it is advised to withdraw potentially interfering medication at least 2 weeks before the testing [42,43].

#### 3.1.2. Chromogranin B and Pancreastatin

Other granins such as CgB have also been researched as potential biomarkers, however their testing availability and, therefore, their clinical usefulness is limited [8]. Pancreastatin—a product of enzymatic cleavage of CgA—has shown to retain similar sensitivity and specificity to CgA, while being unaffected by PPI treatment [44,45]. Elevated concentrations of pancreastatin correlate with a shorter progression-free survival (PFS) and the overall survival (OS) of patients with pancreatic and small bowel NETs, which makes it a potential prognostic biomarker [46]. It seems to be especially useful in metastatic disease and recent data suggests that it compares better to CgA in detecting the progression of midgut NETs [47,48].

### 3.2. 5-Hydroxyindoleacetic Acid

5-hydroxyindoleacetic acid (5-HIAA), a serotonin metabolite, has been one of the longest used biomarkers in neuroendocrine tumors, since 1956 [49]. Serotonin is produced by enterochromaffin cells, most commonly located in the small intestine. It serves a purpose in regulating gastrointestinal tract motility [50]. Elevated levels of serotonin can be observed in neuroendocrine tumors, most commonly of midgut origin. Serotonin-secreting neuroendocrine tumors manifest as carcinoid syndrome. Originally, the term carcinoid was invented by Oberndorfer in 1907 and has been used to describe all NETs [51]. Currently use of the term “carcinoid” is discouraged, due to the confusing terminology applied to it over the years. Serotonin is produced by 70% of all neuroendocrine tumors, with the percentage of serotonin positive gastric and pulmonary NETs reaching as low as 10–35%. Monitoring serotonin itself is challenging, due to fluctuations in its secretion over time as well as differences between individuals, therefore its metabolites, such as 5-HIAA, are preferred [52]. 5-HIAA can be measured both in serum and urine, although the latter is more broadly used. Urine samples need to be collected over a 24 h period, protected from light and added with an acidic compound to ensure stability [53]. The sensitivity of 5-HIAA in diagnosis and monitoring is quite low, around 35%, and strongly depends on serotonin secretion [54]. 5-HIAA urine concentration has shown a positive correlation with the severity of carcinoid syndrome [55]. Higher values are also observed in patients with metastatic midgut NETs, compared to non-metastatic patients, notably with liver metastases. Moreover, 5-HIAA could be a marker of a biochemical response to somatostatin analog treatment and may be useful in the early detection of recurrence post-surgery [21]. Despite a specificity of up to 100% in some trials [25], there are several factors limiting 5-HIAA use in daily clinical practice. Tryptophan-rich food, such as peanuts, bananas, chocolate, coffee and tea, as well as certain medication (e.g., diazepam and phenobarbital), might lead to false-positive results, therefore patients undergoing tests need to adhere to dietary restrictions [56]. In addition, 24 h urine collection is impractical, when compared to liquid biopsy due to a prolonged testing period and the requirement of additional equipment and preparation. The clinical usefulness of 5-HIAA is restricted to serotonin-producing tumors (i.e., manifesting as carcinoid syndrome), which applies to just a fraction of neoplasms. 

### 3.3. Pancreatic Polypeptide, Neuropeptide Y and Peptide YY

Another circulating biomarker, described in literature as secreted by an NEN, is pancreatic polypeptide (PP). It belongs to the same group of peptides as Peptide YY (PYY) and neuropeptide Y (NPY). PP is a 36-amino-acid molecule involved in the regulation of the digestive tract function and food metabolism (i.e., increasing hepatic insulin sensitivity) [57]. Used on its own, PP has a low sensitivity of 41–63% for pNET and 18–53% for gastrointestinal NET [58]. Higher levels are associated with pancreatic tumors and metastatic disease. When used together with CgA, the test can detect NEN with a sensitivity of 84–96% [59]. Peptide YY is very similar to PP, with 18 of its 36 amino acids located in the same positions. PYY cells were found in gastrointestinal NEN tissue, most commonly in rectal NEN, where its presence has been associated with a worse prognosis [60,61]. The data on its use as a circulating biomarker are lacking. Another 36-amino-acid-long peptide is Neuropeptide Y. The elevated plasma levels of NPY have mostly been the focus of research in pheochromocytomas, neuroblastomas and gangliomas [62,63]. In one study by Allen et al., elevated levels of NPY were present in 6 out of 22 gastrointestinal NETs [64]. Whereas PP has some potential applications as a circulating NET biomarker, the utility of PYY and NPY is limited.

### 3.4. Neuron-Specific Enolase

In 1965, Moore and McGregor discovered a protein currently known as neuron-specific enolase (NSE) [65]. NSE is a glycolytic enzyme present in neurons and neuroendocrine cells in the central and peripheral nervous system. Elevated concentrations of NSE in body fluids can be found not only in septic shock and post-traumatic states, but also in conditions associated with cell proliferation, such as neoplasms [66,67]. The latter property was hoped to be useful in detecting NETs, however, research shows that NSE is elevated in just 19% of G1NET and 54% of G2NET cases. Therefore, it seems to be unreliable as a single diagnostic biomarker for well-differentiated tumors, however it can be of added value to CgA in G2NET cases [8,68]. NSE concentrations are significantly higher in NECs with a sensitivity of 63% in large cell neuroendocrine carcinoma (LCNEC) and 62% in small cell neuroendocrine carcinoma (SCNEC) [68]. Moreover, NSE may be useful as a predictor of long-term survival in NEN cases and, thanks to its dependence on cell turnover, it is associated with malignant forms with a higher grading [69].

### 3.5. NETest

In 2007, a National Cancer Institute summit meeting on NETs was held. It was deemed that the currently available biomarkers have severe limitations and it is crucial to develop universal biomarkers for early diagnosis [70]. It has been widely discussed that molecular methods might describe an entity as dynamic and diverse as an NET in a much more adequate way than a single substance [71]. Therefore, in the last couple of years, researchers started moving towards complex multianalyte assays that utilize statistical algorithms.

An NETest is an example of one such method. It is based on evaluating a tumor’s gene expression, i.e., its “biological signature”. After performing a liquid biopsy and isolating the mRNA (messenger RNA), the cDNA (complementary DNA) is synthesized. Subsequently, PCR and gene analysis is performed, and the results are subjected to machine-learning algorithms. The resulting score is given on a scale from 0 to 100% (the normal score cut-off is 20%) [72].

The NETest has shown excellent diagnostic parameters in multiple trials, with both sensitivity and specificity exceeding 90% [73,74,75,76]. In the multicenter study published in 2021, Modlin et al. analyzed two cohorts of patients over 5 years. The first group focused on the NETest evaluation and was made up of 1684 NETs compared with 731 controls, whereas the second group was comparing an NETest with CgA and comprised 922 NETs versus 348 controls. In the described setting, the NETest identified 98% pheochromocytomas, 94% siNET, 91%panNET, 88%bpNET, 80% gastric NET and 79% NETs of the appendix. The NETest was more effective in diagnosing and monitoring NETs than CgA [77]. In a different trial, an NETest was able to detect progression 1 year before imaging methods [78]. Unlike CgA, factors such as PPI treatment and gastritis have no bearing on the results [79].

Overall, the NETest fits the criteria of an optimal biomarker thanks to its outstanding diagnostic properties, prognostic and predictive value that outperforms traditional analytes [80,81,82,83]. Among largely promising results, the NETest too has some potential downsides. Its cost-effectiveness is relatively unknown, and the question remains whether it can be widely introduced. On top of that, there are very few laboratories that are able to perform NETest analysis (i.e., Wren Laboratories in the USA and Sarah Cannon Molecular Diagnostics in Great Britain) [83].

### 3.6. microRNA

microRNAs (miRNA) are a group of small (22 nucleotides in length on average), noncoding RNA molecules that promote or suppress posttranscriptional gene expression [84]. Despite being discovered in 1993, their clinical applications only started gaining traction in the last few years [85]. miRNAs can be identified both in solid tissue as well as in body fluids (inter alia plasma, serum, saliva, CSF and urine). They can be secreted in autocrine, paracrine and endocrine ways (although the exact mechanisms are unknown) [86]. Such properties allow for an identification using a liquid biopsy and potentially making them useful as disease biomarkers [87]. Altered miRNA levels in body fluids are associated with numerous diseases (cardiovascular, gastrointestinal, renal, psychiatric, neoplasms etc.) [88,89,90]. In cancer, miRNAs can promote metastases, regulate angiogenesis and cell metabolism, as well as influence immune evasion and the response to certain treatment methods [87]. It is clear that miRNA dysregulation plays a crucial role in carcinogenesis and understanding the processes behind it might improve diagnosis and the treatment of oncological patients in the future [91].

miRNA have been extensively researched in most common neoplasms, e.g., ovarian cancer, lung cancer and colorectal cancer [92,93,94]. In comparison, little is known about circulating miRNA in NETs. The altered expression of over 100 miRNAs have been described in NETs [95]. So far, no universal target molecule for NETs has been identified, possibly due to the diverse nature of neuroendocrine neoplasms and the fact that many miRNAs are tissue-specific [96]. Moreover, different molecules seem to be expressed in blood and tumor tissue, although some can be detected in both compartments [97]. Malczewska et al. summarized in their systematic review that in panNETs, miR-1290 is absent in tumor tissue, while miR-21 and MiR-133a seem to be present in both. In siNET, miR-7-5p, miR-31, miR-96, miR-133a, miR-182, miR-183, miR-196a and miR-215 can be traced in both blood and tumor tissue, while circulating miRNAs include additionally miR-21, miR-22, miR-150, miR-200a, miR-21, miR-133a and miR-144. Only miR-21 and miR-133a have been described as circulating miRNAs in both locations (the former also presents in lungs) [98].

Li et al. analyzed over 700 circulating miRNAs aiming to differentiate pancreatic cancer from NETs and benign pancreatic conditions. In that setting, the expression of miR-1290 was higher in the pancreatic cancer group vs. the NET group (81% sensitivity and 69% specificity), although no comparison has been made between NETs and other conditions. Several other miR showed statistically significant results (miR-628-3p, miR-550 and miR-1825), however, their diagnostic parameters were of lower value than miR-1290 [99]. Additionally, miR-375-3p distinguishes a low-grade lung NET from non-neuroendocrine lung tumors showing over 90% sensitivity and specificity [100]. miR-375 and miR-133a have been discussed as a biomarker of patient survival due to the down-regulation in tumor metastases of siNET, however, both as tumor tissue biomarkers) [101,102]. miR-375 seems to be particularly interesting, as it has been localized in enteroendocrine cells and has been described as an endocrine system modulator and marker of neuroendocrine differentiation [103,104]. miR-29b is a member of the miR-29 family, which has been researched as a biomarker for several cancers, including lung and ovary [105]. Özdirik et al. described a correlation between miR-29b and CgA levels, though no relation to OS has been shown [106]. Recently, the overexpression of 13 selected circulating miRNAs has been described in NENs and medullar, in comparison to healthy subjects. It was the first study in which a set of circulating miRNAs was identified that could represent a tumor signature for NEN diagnostics [107].

An expert consensus suggests that circulating miRNAs will be of use as a NET biomarker. However, as with most multianalytes, due to their complex nature, any potential tests will have to be based on mathematical algorithms in order to make them clinically viable [49]. A recent study by Nanayakkara et al. described a machine-learning algorithm utilizing a panel of 17 miRNAs that determines 15 NEN types with 98% accuracy. With further research, more refined algorithms will become available [96]. Another problem limiting potential clinical applications has been the unknown influence of treatment on miRNA expression. Somatostatin analogs change the patterns of circulating miRNA; the exact mechanisms of that process are poorly understood [108,109].

### 3.7. Circulating Tumor Cells

In neoplasms, as the tumor grows, certain cells split away from the lesion and enter circulation. These circulating tumor cells (CTC), if certain conditions are met, can settle down in a new location and form metastases [110]. The phenomenon has been described in the 19th century already by Thomas Ashworth, however, it took over 100 years until researchers began to understand the process behind it [111]. The first trials that focused on the isolation and identification of these cells were conducted in the late 20th century, and in 2004 CellSearch was approved by the United States Food and Drug Administration (FDA) as the first device for CTC analysis (at the time for use in breast cancer) [112]. Since then, multiple technologies were developed for detection in peripheral blood, utilizing CTC’s distinct physical properties, immunoaffinity or direct analysis with fiber-optic arrays [113].

In NETs, CTC were detected for the first time in 2011. In a study published by Khan et al., 21% of panNETs and 43% of midgut NETs had detectable CTC. It is important to note however, that all subjects had metastatic disease at the time of the analysis [114]. In the 2013 follow-up study, 49% of patients in the group of 176 had at least one detectable CTC and the association between the presence of CTC and shorter PFS and OS has been described [115]. Additionally, in 2016, research of 138 metastatic NET patients (primary sites included bronchopulmonary pancreas, midgut, hindgut and unknown primary location) was published. A low CTC count and CTC decrease post treatment had a favorable prognosis over a high CTC count, which correlated with a shorter OS [116]. Similar observations have been published in 2019 by Hsieh et al. in the study of Asian NET patients [117]. An effect of CTC presence on the effectiveness of somatostatin analog (SSA) treatment has been evaluated in the CALM-NET trial; patients with no detectable CTC might be more likely to respond positively to the treatment [118].

Despite promising results, studies mentioned above have certain limitations. The patients included have been diagnosed with NETs of different primary locations (foregut, midgut and hindgut tumors), therefore their biological features might differ. Moreover, different treatment methods (SSA included) might have affected the CTC expression and, therefore, the results [119]. The biomarker issue aside, recent findings suggest that a qualitative and quantitative assessment of CTC may be equally important. Mutations present in CTC reflect the genomic aberrations found in tumor tissue, making liquid biopsy a useful option in cases where standard biopsy might not be possible or for tracking changes in a tumor’s genomic landscape. Monitoring these changes can also be useful in establishing mechanisms of resistance to certain forms of treatment [120].

NETs are generally indolent tumors; about a fifth of the patients present with metastases at diagnosis [121]. In mouse models of aggressive tumors, such as breast or pancreatic cancer, CTC have been detected even at the early stage of the disease [122]. However, the question remains whether the same can be applied to NETs given CTC’s limited sensitivity in tumors with more metastatic potential. What is more, CTC’s potential uses as prognostic or predictive biomarkers require further research. With the lack of a large cohort, multicenter studies remain an important unmet need.

### 3.8. Circulating Tumor DNA and Cell-Free DNA

Circulating tumor DNA (ctDNA) and cell-free DNA (cfDNA) are a novel tool that can be used to describe NETs molecular features. Whereas ctDNA are fragments of DNA derived from a tumor and found in the circulation, cfDNA are a broader term and also include fragments of nucleic acid that do not originate in a tumor. The main source of circulating tumor DNA seems to be the apoptosis, although the exact mechanism of releasing ctDNA into the body fluids remains unclear [123]. The principle behind this test is the identification of circulating DNA and its molecular rearrangements, which may affect treatment choices [124].

The presence of ctDNA was first reported in 1948 by Mandel and Metais, who detected cell-free nucleic acids in the blood of cancer patients [125]. Since then, ctDNA has been widely studied as an alternative for tissue biopsies in malignancies, however, the data about its use in NETs remains scarce. A relative lack of known, unique to NEN, neoplasm-promoting mutations presents a significant limitation for the use of ctDNA [126,127].

Some of the upsides of circulating nucleic acid analysis include the simplicity of obtaining the material and minimally invasive monitoring of the tumor during therapy by liquid biopsy. The risk of false negative results seems to be the main limitation of this method, due to variable amounts of DNA that tumors may release into circulation [124,128].

It has been reported that the presence of ctDNA in body fluids is linked to the localization of the primary tumor and metastatic lesions [129,130,131,132]. Oversoe et al. described elevated levels of cfDNA in panNET and siNET patients compared to healthy controls [133]. Tumors with liver metastases and a high proliferative index and necrosis, features which are often characteristic of NEC, are associated with a high ctDNA concentration [131]. Boons et al., described a correlation between the presence of ctDNA and a higher grading [132]. On the contrary to NECs, NETs (which are generally slow growing tumors) have a lower cell loss index and ctDNA release and can often be ctDNA negative [130]. Quantitative analysis of ctDNA may also be useful to assess tumor volume and in an early diagnosis of relapse after surgery and as a predictive factor of response to treatment [134,135,136,137]. OS and PFS appear to be significantly worse in ctDNA-positive than in ctDNA-negative patients [130].

Another important aspect of cfDNA analysis is the possibility of methylation pattern analysis. Abnormal distribution of DNA methylation has been described in early carcinogenesis and may be helpful in the detection, monitoring and treatment response prediction [138]. A number of studies have been performed describing in NET tumor tissue methylation patterns compared to healthy controls [139,140,141,142]. Mettler et al. analyzed the cfDNA characteristics of 63 NEN patients in comparison with healthy controls. A higher cfDNA concentration and hypomethylation patterns have been found in advanced NEN and their association with tumor burden and a worse prognosis has been described [143]. 

Although the sensitivity of ctDNA may be lower than the currently used analytes, it is a highly specific biomarker, which can be especially useful in rare diseases [130,144]. Moreover, in the qualitative analysis of ctDNA, both copy number alterations and point mutations in DNA is clinically relevant, namely for screening patients who are eligible for targeted therapies. However, the application of ctDNA in NETs requires further study [145].

### 3.9. Other Potential Biomarkers and 19th Annual Enets Conference Abstracts

Some other areas of interest in the field of NET biomarkers have been described in recent years that are not covered in detail by this review, e.g., long non-coding RNA and tumor-infiltrating platelets. However, the data on these remains scarce [146,147]. In a recent report, Hinterleitner et al. described elevated levels of platelet-expressed synaptophysin (pSyn) in NEN compared to healthy donors. A high expression of pSyn was shown to correlate with a shorter PFS, higher tumor stages, the presence of metastases and a higher tumor proliferation rate [148].

The 19th Annual ENETS Conference took place in March 2022. Some of the research presented during the conference focused on NEN biomarker development. La Salvia et al. presented an analysis of extra-pancreatic NETs metabolomics profile, some of which can be used as independent prognostic biomarkers. Some of the findings have already been published in a peer-reviewed journal [149,150]. Another interesting finding has been the analysis of Copy Number Alterations (CNAs) in cfDNA. The method utilizes whole-genome sequencing of cfDNA (its ctDNA fraction, precisely) in material acquired by liquid biopsy. CNAs found in analyzed material showed a sensitivity and specificity for NENs of 62% and 86%, respectively [151]. Garcia Alvarez et al. analyzed the plasma of panNETs and giNETs prior to the start of Lenvatinib. High levels of angiopoietin 2 (Ang2) and low levels of fibroblast growth factor 2 (FGF-2) resulted in a better response to treatment, which may point to them being useful as predictive biomarkers [152]. One study focused on ctDNA in NEN and its clinical utility for monitoring. The lack of identifiable ctDNA in patients with stable disease has been described, which may help in selecting a group of patients with no need for intensive monitoring [153]. Serum Activin A has been researched as an alternative to NT-proBNP in CHD patients, however, its diagnostic parameters for the detection of CHD have been subpar [154]. Schalin-Jantti et al. presented an analysis of clinical factors (CF) and novel plasma proteins (NPP) in G1 and G2 SI-NET patients using data mining and machine learning methods. The study focused on establishing a multi biomarker strategy for NET. The combination of CF and NPP allowed for the identification of stable and progressive disease subgroups [155]. This research is yet another example of how useful machine learning might be in advancing patient care. Finally, a study focusing on an NETest have been presented by van Treijen et al. showing its function in predicting treatment response and individualizing treatment decision [156]. The latter conclusion is especially important as the individualization of therapy has been a major talking point during the 19th Annual ENETS Conference. 

## 4. Conclusions

Over the years, multiple NET biomarkers have been researched, developed and used. From simple substances secreted by the tumor to complex mathematical algorithms, there is a wide range of biomarkers to choose from. Despite this, there is still an unmet need for the development of widely available and accurate NET biomarkers. Experts specializing in NETs agree that the currently used analytes have several limitations and that multianalyte panels based on the genetic signature of the tumor should be the course of future research [49]. Describing different aspects of a disease as complex and heterogeneous as NET based on a single substance is insufficient. In comparison, utilizing mathematical algorithms allows for a more comprehensive depiction of the state of the disease (thanks to the numerous variables that are included, instead of just a single one) [157]. In a recent study, Kidd et al. described the potential expansion of the NETest, improving its statistical parameters even further [158]. This is yet another advantage of machine learning algorithms; With new discoveries, they can be tweaked for even more accurate analysis. A question often raised is the cost effectiveness of the new biomarkers [159]. Measuring a single substance is markedly less expensive than molecular tests, however, a more efficient biomarker will allow for a decreased spending on imaging and treatment [80].

As shown by this review, there is still room for improvement in the field of NET biomarkers. A number of analytes, such as miRNA, CTC and NETest have shown promising results, however, their use in daily clinical practice is currently limited by either their low availability or lack of standardization.

Out of the potential biomarkers mentioned above, the NETest offers superior diagnostic parameters compared to traditional analytes and has been shown to detect progression and disease recurrence quicker than imaging methods. It is also useful in the assessment of the response to radioisotope treatment and radicality of surgical intervention. As stated in the recently published guidelines of the Polish Network of Neuroendocrine Tumours, the use of an NETest in everyday clinical practice will enable the optimal inclusion of the test in the management algorithms in the Polish population of patients with NEN [8].

However, with the NETest limited availability, there is still place for traditional analytes. In accordance with the updated guidelines of the Polish Network of Neuroendocrine Tumours, we advise utilizing CgA for monitoring during treatment and as a prognostic biomarker in colorectal NEN [8,160]. In small intestine and pancreatic NEN, measuring CgA has a utility before introducing treatment and for monitoring. Additionally, in patients diagnosed with small intestine NEN, bronchopulmonary NEN or when suspecting carcinoid syndrome, it is recommended to measure 5-HIAA in urine (at least two samples, collected over 24 h period each) [8,161,162]. Though not a NET biomarker sensu stricto, the N-terminal prohormone of brain natriuretic peptide (NT-proBNP) should be measured for the diagnosis and monitoring of carcinoid heart disease in carcinoid syndrome patients [163]. In patients presenting with symptoms characteristic of functional NETs, we recommend measuring the hormones linked with the suspected syndrome (as shown on Table 2). As discussed earlier in this review, medical decisions shouldn’t be taken solely on the basis of change in biomarker concentration, due to their several limitations.

To summarize, despite the recent advances in the field of NET biomarkers, novel analytes have not yet been introduced into wider use. Some of them (such as an NETest) show a lot of promise and with a wider availability, they offer a significant improvement over traditional analytes. Until they become a routine tool in NET diagnostics, biomarkers such as CgA and 5-HIAA can still be a helpful option in select cases.

## Figures and Tables

**Table 1 jcm-11-05542-t001:** Examples of neuroendocrine tumors’ primary locations of different embryonic origin.

Foregut	Midgut	Hindgut
Thymus	Jejunum	Distal 1/3 of transverse colon
Esophagus	Appendix	Descending colon
Bronchus	Ileum	Sigmoid colon
Lung	Ascending colon	Rectum
Stomach	Proximal 2/3 of transverse colon	
Pancreas		
Duodenum		

**Table 2 jcm-11-05542-t002:** Functional pancreatic NET and corresponding specific biomarkers.

Type of Tumor	Secreted Hormone	Incidence (New/100,000/Year) [9]
Insulinoma	Insulin	1–32
Gastrinoma	Gastrin	0.5–21.5
VIPoma	Vasoactive Intestinal Peptide	0.05–0.2
Glucagonoma	Glucagon	0.01–0.1
Somatostatinoma	Somatostatin	Rare < 0.1
GRHoma	GH-releasing hormone	Rare
Ghrelinoma	Ghrelin	Unknown (>100 cases described)
ACTHoma	ACTH	Rare
Pancreatic NET causing Carcinoid Syndrome	Serotonin	Rare (<100 cases)
Pancreatic NET causing hypercalcemia	PTHrP (Parathyroid Hormone-related Peptide)	Rare

## Data Availability

Not applicable.

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
