# Peer review of "Circulating Neuroendocrine Tumor Biomarkers: Past, Present and Future"

_jcm, 2022, doi:10.3390/jcm11195542_

Round 1

Reviewer 1 Report

Clinical utility and application of neuroendocrine tumors biomarkers is relatively important topic due to lack of universal biomarkers for NET management.

Authors summarized results of the most clinically important older and newer NETs biomarkers what is the strength of the work, however there is several issues which should be considered before manuscript publication especially it is a review article.

Introduction:

Could you explain the importace of NET deviation showed in table 1  in the context of biomarkers?

-         Could you provide in table 2 the frequencies of functional NETs, set them in order of incidence?

I can not find chapter results which shows how many researches described in materials ad methods section have you found and how you use/compare their results

Discussion:

-    -      5-HIAH – you provide the information about sensitivity of HIAA measurements – are they connected to diagnosis? disease progression? recurrence?

-       -   CgA -  you stress that depending on the type of the tumor and location, sensitivity and  specificity range between 68-81% and 56-100% respectively. Similarly to 5- 126 HIAA, its sensitivity and specificity differs depending on the location - midgut tumors express CgA most often, foregut and hindgut less so. Could you provide information how robust CgA values are in different localizations to show in which clinical application they really valuable?

-    -      Neuron-specific enolase – you wrote it can however be of added value to chromogranin A in poorly-differentiated tumors as its concentrations are significantly higher - could you provide which differences in Neuron-specific enolase concentrations which are clinically significant

-        There is no explanation why you omit other potentially useful biomarkers as e.g. pancreatic polypeptide or peptide YY

Could you sum up your findings with the information which biomarker, in your opinion (based on particular biomarkers general availability) is best clinical option for NET diagnosis, progression assessment and recurrence detection?

Author Response

Dear Reviewer, 

we appreciate taking your time to review the manuscript that we have submitted. In the file attached you can find our response to the first round of peer-review.

Kind regards, Paweł Komarnicki

Reviewer 2 Report

The search for reliable biomarker is a urgent need in the field of neuroendocrine neoplasms, thus the topic is interesting. Nevertheless there are some major points that need to be addressed.

The distinction based on embryonic origin does not take into account rare sites of origin and the WHO classifications should be reported to highlight the importance of differentiation in diagnosis, management and prognosis.

In materials and methods, it is reported among search terms “NSE”, which is a biomarker in less differentiated NET and neuroendocrine carcinoma (NEC). Consequently if both NET and NEC are included in the rewiew, I suggest to change the title into “Neuroendocrine Neoplasms Biomarkers”. Furthermore, the terms “neuroendocrine neoplasm” (included among keywords) and NEC should have been included in the search; please explain this discrepancy.

Line 42 Limits in using the hormones released by functional NEN as biomarkers should be added (reliability, availability…)

Lines 85-86 According to the WHO classification the use of term “carcinoid” is currently discouraged in midgut NET since differences in terminology and classifications criteria have caused considerable confusion, please correct.

I suggest to begin the description of biomarkers with granins, as CgA is the most broadly used biomarker. Additionally, as stated in NCCN guidelines, it should be specified that CgA should not be relied upon in isolation as a diagnostic test. Furthermore, a paragraph specifically regarding the future of biomarker in NEN (as stated in the title) should be added before conclusions.

When discussing the hormonal workup, it should be stated that it should be guided by the presence of symptoms of the excess hormone as screening for hormones in asymptomatic individuals is not routinely required, according to NCCN guidelines (NCCN Guidelines Version 1.2021).

The description of NSE needs to be detailed (Oberg K et al. ENETS Consensus Guidelines for Standard of Care in Neuroendocrine Tumours: Biochemical Markers. Neuroendocrinology. 2017;105(3):201-211. doi: 10.1159/000472254).

With regard to MiRNAS relevance in differential diagnosis and prognosis please consider citing the following paper by Colao A et al. Clinical Epigenetics of Neuroendocrine Tumors: The Road Ahead. Front Endocrinol (Lausanne). 2020 Dec 15;11:604341. doi: 10.3389/fendo.2020.604341. Moreover, a set of circulating miRNAs has been recently selected as possible diagnostic/prognostic biomarkers of NEN, as described in a paper which is worth citing by Melone V et al. (Identification of functional pathways and molecular signatures in neuroendocrine neoplasms by multi-omics analysis. J Transl Med. 2022 Jul 6;20(1):306. doi: 10.1186/s12967-022-03511-7).

Author Response

(The authors gave the same response as above.)

Reviewer 3 Report

Although the authors' effort is worthy, the article lacks the originality required to be published in this journal. In addition, the English should be revised. Sometimes there are phrases that are not very understandable or not good for a scientific article (the use of acronyms should be revised, contractions, etc.). 

Author Response

(The authors gave the same response as above.)

Reviewer 4 Report

This is a nice review manuscript describing a number of well-known and new biomarkers in neuroendocrine tumors. I have a few remarks.

1. The current WHO classification describes neuroendocrine neoplasms (NEN), which include neuroendocrine tumors (NET) and neuroendocrine cancers (NEC). The authors should acknowledge this system and apply it to their work.

2.  The choice of biomarkers seems rather peculiar: synaptophysin as one important serological and histopathology marker isnt mentioned at all, the same goes for somatostatin receptors and other markers. The review should at least aim to cover the most relevant markers.

3. The manuscript completely misses imaging biomarkers, which play an important role in many treatment decisions. This topic area should be covered, including targets of recently described imaginng biomarkers like GLP-1R and GIPR.

Author Response

(The authors gave the same response as above.)

Round 2

Reviewer 1 Report

The authors have significantly improved the manuscript.

One suggestion in regard to the discussion:  you have suggested NETest as the most valuable biomarker, which I completely agree with, but unfortunately its availability is very limited so its current recommendation as optimal NET biomarker does not seem clinically appropriate.  Could you summarize (in 2-3 sentences) your suggestions for the currently commonly available biomarkers in the different NET subtypes (as the best clinical option for NET diagnosis, assessment of progression and detection of NET recurrence), e.g. according to the deviations shown in Table 1?

Author Response

Dear Reviewer,

Thank you for your comments and suggestion in second round of peer-review. We have responded to them in the attached file.

Kind regards, Paweł Komarnicki

Reviewer 3 Report

The review article still, in my opinion, lacks the minimum interest to be published in this Journal. It focuses on extremely well known serum markers in NETs. In contrast, there is little or no development in the study of new NET-focused platforms, such as NEtest. serum methylation profiling, ctDNA, etc. In review articles of this type, the most up-to-date evidence focuses on those recent data presented at recent congresses, although not yet published, so the search methodology can also be improved. 

In summ, I consider that the article has not improved enough in this second draft to be published. 

Author Response

Dear Reviewer,

Thank you for your comments and suggestion in second round of peer-review. You can find our comments in the attached file.

Kind regards, Paweł Komarnicki

Reviewer 4 Report

The manuscript is now acceptable for publication.

Author Response

Dear Reviewer,

thank you for your kind suggestions. We have slightly modified the manuscript according to other reviewer's suggestions. We have added 1 paragraph describing circulating tumor DNA and cell-free DNA as well as discussed latest finding presented during 2022 ENETS Conference.

Kind regards, Paweł Komarnicki